# LET'S VERIFY STEP BY STEP

**Hunter Lightman**\*, **Vineet Kosaraju**\*, **Yura Burda**\*, **Harri Edwards, Bowen Baker,**
**Teddy Lee, Jan Leike, John Schulman, Ilya Sutskever & Karl Cobbe**\*
OpenAI
San Francisco, CA, USA
`karl@openai.com`

## ABSTRACT

In recent years, large language models have greatly improved in their ability to perform complex multi-step reasoning. However, even state-of-the-art models still regularly produce logical mistakes. To train more reliable models, we can turn either to outcome supervision, which provides feedback for a final result, or process supervision, which provides feedback for each intermediate reasoning step. Given the importance of training reliable models, and given the high cost of human feedback, it is important to carefully compare the both methods. Recent work has already begun this comparison, but many questions still remain. We conduct our own investigation, finding that process supervision significantly outperforms outcome supervision for training models to solve problems from the challenging MATH dataset. Our process-supervised model solves 78% of problems from a representative subset of the MATH test set. Additionally, we show that active learning significantly improves the efficacy of process supervision. To support related research, we also release PRM800K, the complete dataset of 800,000 step-level human feedback labels used to train our best reward model.

## 1 INTRODUCTION

Large language models are capable of solving tasks that require complex multi-step reasoning by generating solutions in a step-by-step chain-of-thought format (Nye et al., 2021; Wei et al., 2022; Kojima et al., 2022). However, even state-of-the-art models are prone to producing falsehoods — they exhibit a tendency to invent facts in moments of uncertainty (Bubeck et al., 2023). These *hallucinations* (Maynez et al., 2020) are particularly problematic in domains that require multi-step reasoning, since a single logical error is enough to derail a much larger solution. Detecting and mitigating hallucinations is essential to improve reasoning capabilities.

One effective method involves training reward models to discriminate between desirable and undesirable outputs. The reward model can then be used in a reinforcement learning pipeline (Ziegler et al., 2019; Stiennon et al., 2020; Nakano et al., 2021; Ouyang et al., 2022) or to perform search via rejection sampling (Nichols et al., 2020; Shen et al., 2021; Cobbe et al., 2021). While these techniques are useful, the resulting system is only as reliable as the reward model itself. It is therefore important that we study how to most effectively train reliable reward models.

In closely related work, Uesato et al. (2022) describe two distinct methods for training reward models: outcome supervision and process supervision. Outcome-supervised reward models (ORMs) are trained using only the final result of the model's chain-of-thought, while process-supervised reward models (PRMs) receive feedback for each step in the chain-of-thought. There are compelling reasons to favor process supervision. It provides more precise feedback, since it specifies the exact location of any errors that occur. It also has several advantages relevant to AI alignment: it is easier for humans to interpret, and it more directly rewards models for following a human-endorsed chain-of-thought. Within the domain of logical reasoning, models trained with outcome supervision regularly use incorrect reasoning to reach the correct final answer (Zelikman et al., 2022; Creswell et al., 2022). Process supervision has been shown to mitigate this misaligned behavior (Uesato et al., 2022).

---

\*Primary authors. Correspondence to: Karl Cobbe <karl@openai.com>

Despite these advantages, Uesato et al. (2022) found that outcome supervision and process supervision led to similar final performance in the domain of grade school math. We conduct our own detailed comparison of outcome and process supervision, with three main differences: we use a more capable base model, we use significantly more human feedback, and we train and test on the more challenging MATH dataset (Hendrycks et al., 2021).

Our main contributions are as follows:

1. We show that process supervision can train much more reliable reward models than outcome supervision. We use our state-of-the-art PRM to solve 78.2% of problems from a representative subset of the MATH test set.

2. We show that a large reward model can reliably approximate human supervision for smaller reward models, and that it can be used to efficiently conduct large-scale data collection ablations.

3. We show that active learning leads to a $2.6\times$ improvement in the data efficiency of process supervision.

4. We release our full process supervision dataset, PRM800K, to promote related research.

## 2 METHODS

We perform a comparison of outcome and process supervision, following a similar methodology to Uesato et al. (2022). Outcome supervision can be provided without humans, since all problems in the MATH dataset have automatically checkable answers. In contrast, there is no simple way to automate process supervision. We therefore rely on human data-labelers to provide process supervision, specifically by labelling the correctness of each step in model-generated solutions.

We conduct experiments in two separate regimes: large-scale and small-scale. Each has its own advantages, and they offer complimentary perspectives. At large-scale, we finetune all models from GPT-4 (OpenAI, 2023). We focus on advancing the state-of-the-art by training the most reliable ORM and PRM possible. Unfortunately the training sets for these reward models are not directly comparable, for reasons we will discuss in Section 3. These models are therefore not ideal for making an apples-to-apples comparison of outcome and process supervision. To address this flaw, we also train models at small-scale, where we can conduct a more direct comparison. In order to remove our dependence on costly human feedback, we use a large-scale model to supervise small-scale model training. This setup enables us to conduct several important ablations that would otherwise be infeasible.

### 2.1 SCOPE

At each model scale, we use a single fixed model to generate all solutions. We call this model the *generator*. We do not attempt to improve the generator with reinforcement learning (RL). When we discuss outcome and process supervision, we are specifically referring to the supervision given to the reward model. We do not discuss any supervision the generator would receive from the reward model if trained with RL. Although finetuning the generator with RL is a natural next step, it is intentionally not the focus of this work.

We instead focus exclusively on how to train the most reliable reward model possible. We evaluate a reward model by its ability to perform best-of-N search over uniformly sampled solutions from the generator. For each test problem we select the solution ranked highest by the reward model, automatically grade it based on its final answer, and report the fraction that are correct. A reward model that is more reliable will select the correct solution more often.

### 2.2 BASE MODELS

All large-scale models are finetuned from the base GPT-4 model (OpenAI, 2023). This model has been pretrained solely to predict the next token; it has not been pretrained with any Reinforcement Learning from Human Feedback (RLHF) (Christiano et al., 2017). The small-scale base models are similar in design to GPT-4, but they were pretrained with roughly 200 times less compute. As

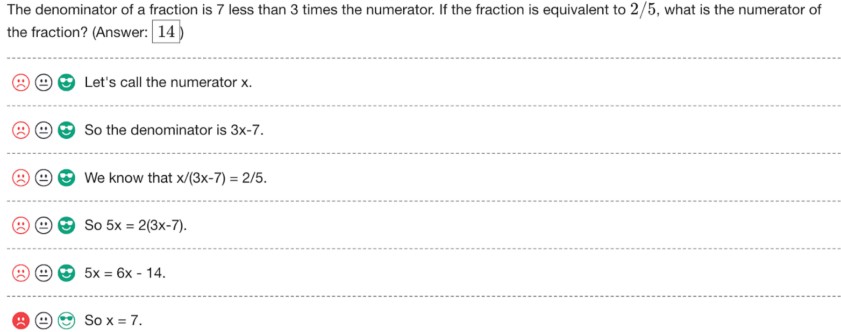

Figure 1: A screenshot of the interface used to collect feedback for each step in a solution.

an additional pretraining step, we finetune all models on a dataset of roughly 1.5B math-relevant tokens, which we call MathMix. Similar to Lewkowycz et al. (2022), we find that this improves the model's mathematical reasoning capabilities. Details on how this dataset was constructed can be found in Appendix A.

## 2.3 GENERATOR

To make parsing individual steps easier, we train the generator to produce solutions in a newline delimited step-by-step format. Specifically, we few-shot generate solutions to MATH training problems, filter to those that reach the correct final answer, and finetune the base model on this dataset for a single epoch. This step is not intended to teach the generator new skills; it is intended only to teach the generator to produce solutions in the desired format.

## 2.4 DATA COLLECTION

To collect process supervision data, we present human data-labelers with step-by-step solutions to MATH problems sampled by the large-scale generator. Their task is to assign each step in the solution a label of *positive*, *negative*, or *neutral*, as shown in Figure 1. A positive label indicates that the step is correct and reasonable. A negative label indicates that the step is either incorrect or unreasonable. A neutral label indicates ambiguity. In practice, a step may be labelled neutral if it is subtly misleading, or if it is a poor suggestion that is technically still valid. We permit neutral labels since this allows us to defer the decision about how to handle ambiguity: at test time, we can treat neutral labels as either positive or negative. A more detailed description of the labelling instructions is provided in Appendix D.

We label solutions exclusively from the large-scale generator in order to maximize the value of our limited human-data resource. We refer to the entire dataset of step-level labels collected as PRM800K. The PRM800K training set contains 800K step-level labels across 75K solutions to 12K problems. To minimize overfitting, we include data from 4.5K MATH test problems in the PRM800K training set, and we therefore evaluate our models only on the remaining 500 MATH test problems. More details about this test set can be found in Appendix C.

During data collection, we must decide which solutions to surface to data-labelers. The most straightforward strategy is to uniformly surface solutions produced by the generator. However, if we surface solutions that make obvious errors, the human feedback we get is less valuable. We would prefer to surface solutions that are more likely to fool our best reward model. To that end, we attempt to strategically select which solutions to show data-labelers. Specifically, we choose to surface *convincing wrong-answer* solutions. We use the term *convincing* to refer to solutions that are rated highly by our current best PRM, and we use *wrong-answer* to refer to solutions that reach an incorrect final answer. We use this slightly verbose phrasing to emphasize the fact that correctness is determined solely by checking the final answer, a process which occasionally leads to misgraded solutions. We expect to gain more information from labeling convincing wrong-answer solutions, since we know the PRM is mistaken about at least one step in each such solution.

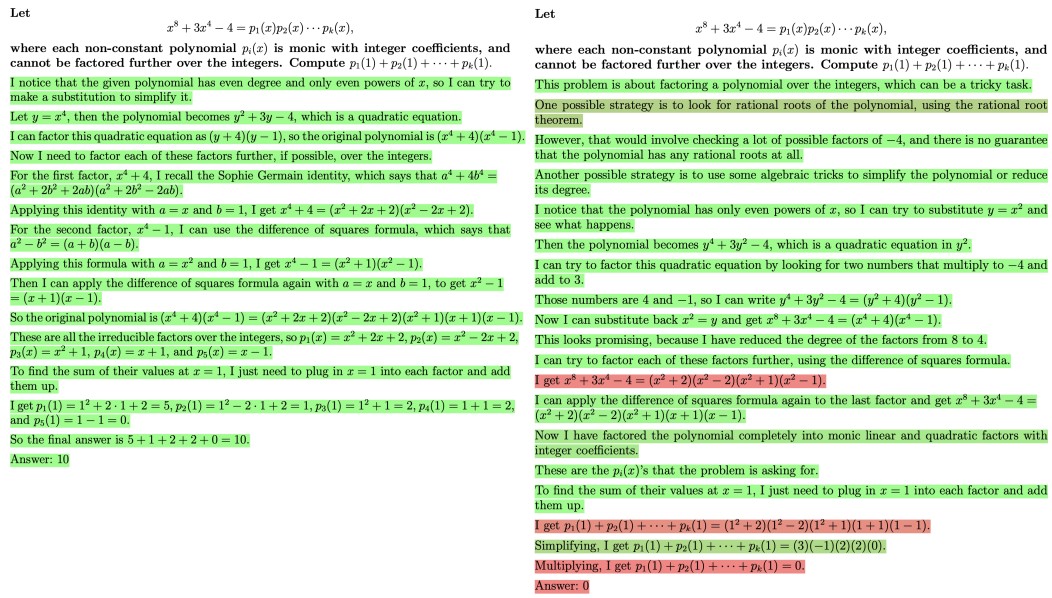

Figure 2: Two solutions to the same problem, graded by the PRM. The solution on the left is correct while the solution on the right is incorrect. A green background indicates a high PRM score, and a red background indicates a low score. The PRM correctly identifies the mistake in the incorrect solution.

In addition to using this selection strategy, we also iteratively re-train our PRM using the latest data at several points in the data collection process. At each iteration, we generate N solutions per problem and surface only the top K most convincing wrong-answer solutions to data-labelers. We experiment with either applying this top-K filtering at a problem level (K solutions per problem) or globally across the dataset (K solutions in total, unequally distributed among problems). Since the data collection process is expensive, it was not feasible to conduct at-scale ablations of these decisions. However, we perform several surrogate ablations in Section 4, using our largest PRM as a labelling oracle for a smaller PRM. More details about data collection can be found in Appendix B.

## 2.5 OUTCOME-SUPERVISED REWARD MODELS (ORMS)

We train ORMs following a similar methodology to Cobbe et al. (2021). We uniformly sample a fixed number of solutions per problem from the generator, and we train the ORM to predict whether each solution is correct or incorrect. In practice, we usually determine correctness by automatically checking the final answer, but in principle these labels could be provided by humans. At test time, we use the ORM's prediction at the final token as the overall score for the solution. We note the automatic grading used to determine ORM targets is not perfectly reliable: *false positives* solutions that reach the correct answer with incorrect reasoning will be misgraded. We discuss additional ORM training details in Appendix E.

## 2.6 PROCESS-SUPERVISED REWARD MODELS (PRMS)

We train PRMs to predict the correctness of each step after the last token in each step. This prediction takes the form of a single token, and we maximize the log-likelihood of these target tokens during training. The PRM can therefore be trained in a standard language model pipeline without any special accommodations. To determine the step-level predictions at test time, it suffices to perform a single PRM forward pass over the whole solution. We visualize large-scale PRM scores for two different solutions in Figure 2. To compare multiple solutions, it is necessary to compute a single score for each solution. This is an important but straightforward detail: we define the PRM score for a solution to be the probability that every step is correct under the PRM. We implement this as the product of the correctness probabilities for each step. We describe other possible scoring strategies and additional PRM training details in Appendix F.

|  | ORM | PRM | Majority Voting |
|---|---|---|---|
| % Solved (Best-of-1860) | 72.4 | **78.2** | 69.6 |

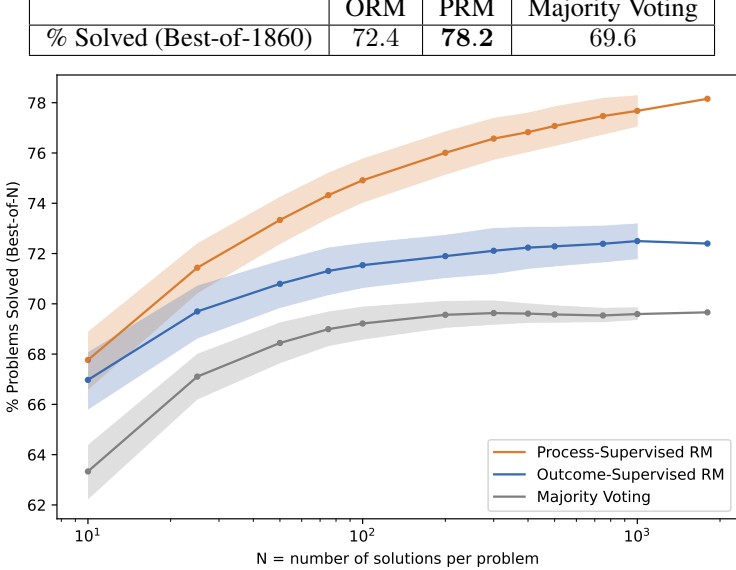

Figure 3: A comparison of outcome-supervised and process-supervised reward models, evaluated by their ability to search over many test solutions. Majority voting is shown as a strong baseline. For $N \leq 1000$, we visualize the variance across many subsamples of the 1860 solutions we generated in total per problem.

When we provide process supervision, we deliberately choose to supervise only up to the first incorrect step. This makes the comparison between outcome and process supervision more straightforward. For correct solutions, both methods provide the same information, namely that every step is correct. For incorrect solutions, both methods reveal the existence of at least one mistake, and process supervision additionally reveals the precise location of that mistake. If we were to provide additional process supervision beyond the first mistake, then process supervision would have an even greater information advantage. This decision also keeps the labelling cost similar for humans: without relying on an easy-to-check final answer, determining the correctness of a solution is equivalent to identifying its first mistake. While most MATH problems do have easy-to-check final answers, we expect this to not remain true in more complex domains.

## 3  LARGE-SCALE SUPERVISION

We train the large-scale PRM using the step-level labels in PRM800K. To ensure the large-scale ORM baseline is as strong as possible, we train on 100 uniform samples per problem from the generator. This means the ORM training set has no overlap with PRM800K, and it is an order of magnitude larger. Although these two training sets are not directly comparable, each represents our best attempt to advance the state-of-the-art with each form of supervision. We note that training the ORM solely on PRM800K solutions would be problematic, since our active learning strategy has heavily biased the dataset towards wrong-answer solutions. We did explore training the ORM on a superset of PRM800K solutions, by mixing in uniformly sampled solutions, but we found that this did not improve ORM performance.

Figure 3 shows how the best-of-N performance of each reward model varies as a function of N. Since majority voting is known to be a strong baseline (Wang et al., 2022; Lewkowycz et al., 2022), we also include this method as a point of comparison. While the ORM performs slightly better than the majority voting baseline, the PRM strongly outperforms both. Not only does the PRM reach higher performance for all values of N, but the performance gap widens as N increases. This indicates that the PRM is more effective than both the ORM and majority voting at searching over a large number of model-generated solutions. We experimented with using RM-weighted voting (Li et al., 2022; Uesato et al., 2022) to combine the benefits of the PRM and majority voting, but this did not noticeably improve performance. We use a specific subset of the MATH test set for evaluation,

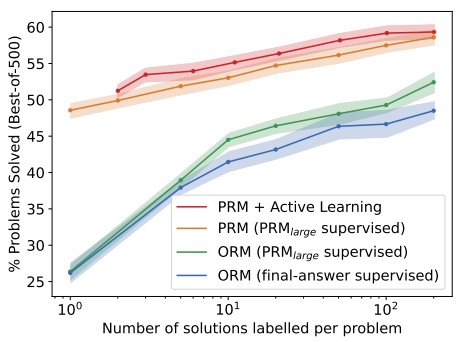
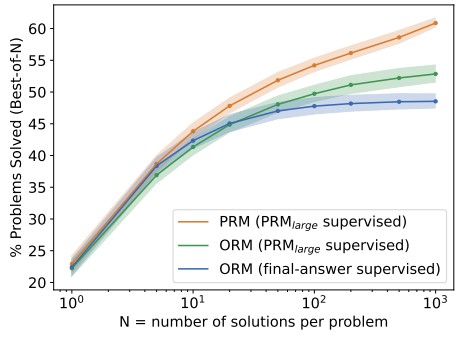

(a) Four series of reward models trained using different data collection strategies, compared across training sets of varying sizes.

(b) Three reward models trained on 200 samples/problem using different forms of supervision, compared across many test-time compute budgets.

Figure 4: A comparison of different forms of outcome and process supervision. Mean and standard deviation is shown across three seeds.

which we describe in Appendix C. We further break down these results by problem difficulty in Appendix G.

## 4 SMALL-SCALE SYNTHETIC SUPERVISION

We find that the PRM outperforms the ORM at large-scale, but this result alone paints an incomplete picture. To better compare outcome and process supervision, there are two confounding factors that must be isolated. First, the training sets for the ORM and the PRM are not directly comparable: the PRM training set was constructed using active learning, is biased towards answer-incorrect solutions, and is an order of magnitude smaller. Second, the final-answer grading will provide positive labels to spurious solutions that reach the correct final answer despite incorrect reasoning. This could damage ORM performance, an effect we may or may not want to attribute to outcome supervision more generally.

Due to the high cost of collecting human feedback, we cannot easily ablate these factors using human labelers. We instead perform the relevant ablations by using the large-scale PRM to supervise smaller models. This setup enables us to simulate a large amount of data collection at a modest cost. For the remainder of this section, we refer to the large-scale PRM from Section 3 as $PRM_{large}$.

### 4.1 PROCESS VS OUTCOME SUPERVISION

We now conduct a direct comparison of outcome and process supervision. We first sample between 1 and 200 solutions per problem from a small-scale generator. For each dataset, we provide three forms of supervision: process supervision from $PRM_{large}$, outcome supervision from $PRM_{large}$, and outcome supervision from final-answer checking. The choice of supervision is the only difference between these three series of reward models, which are otherwise trained on identical datasets. See Appendix H for more details about how $PRM_{large}$ is used for outcome and process supervision.

In Figure 4a, we evaluate each reward model by its best-of-500 selection. We see that process supervision significantly outperforms both forms of outcome supervision at all data collection scales. In Figure 4b, we evaluate the best reward model from each series by its best-of-N performance across different values of N. We see that using $PRM_{large}$ for outcome supervision is noticeably more effective than final-answer checking. This can be explained by the fact that $PRM_{large}$ provides better supervision for solutions that reach the correct final answer using incorrect reasoning.

It is not clear whether supervision by $PRM_{large}$ or by final-answer checking represents the more appropriate outcome supervision baseline. While final-answer supervision is more explicitly outcome based, its main weakness — the existence of false positives — is arguably over-emphasized in the

Table 1: We measure out-of-distribution generalization using recent STEM tests. We evaluate the outcome-supervised RM, the process-supervised RM, and majority voting using 100 test samples per problem.

|  | ORM | PRM | Majority Vote | # Problems |
|---|---|---|---|---|
| AP Calculus | 68.9% | **86.7**% | 80.0% | 45 |
| AP Chemistry | 68.9% | **80.0**% | 71.7% | 60 |
| AP Physics | 77.8% | **86.7**% | 82.2% | 45 |
| AMC10/12 | 49.1% | **53.2**% | 32.8% | 84 |
| Aggregate | 63.8% | **72.9**% | 61.3% | 234 |

MATH dataset. Outcome supervision by $PRM_{large}$ better represents outcome supervision in domains that are less susceptible to false positives. We consider outcome supervision by $PRM_{large}$ to be the more relevant baseline, but we encourage the reader to draw their own conclusions.

## 4.2 ACTIVE LEARNING

Finally, we investigate the impact of active learning. We train a small-scale reward model, $PRM_{selector}$, on a single sample from each problem, and we use this model to score 1000 samples per problem. To train each of our larger reward models, we select $N$ samples per problem such that $80\%$ are the most convincing (according to $PRM_{selector}$) wrong-answer samples, and $20\%$ are the most convincing samples that remain (right- or wrong-answer). We score the selected samples with $PRM_{large}$ and train on those scores. This process ensures that all samples are relatively convincing under $PRM_{selector}$, that a large fraction are known to contain at least one mistake, and that our overall dataset is not too heavily biased toward wrong-answer solutions. Performance of this data labelling scheme is shown in Figure 4a. By comparing the slopes of the line of best fit with and without active learning, we estimate that this form of active learning is approximately 2.6x more data efficient than uniform data labelling. We note that the model trained on the largest active learning dataset (200 samples per problem) appears to slightly underperform the expected trend line. Our best explanation for this observation is that 200 samples represents a significant fraction of the overall selection pool (1000 samples) and that this relative lack of diversity limits the possible upside from active learning.

We also performed a preliminary investigation into the impact of iteratively retraining $PRM_{selector}$ throughout data collection. Between iterations, we re-trained $PRM_{selector}$ using all currently labeled data. Unfortunately, we observed instability in this process which we were unable to diagnose. The resulting reward models performed no better than the models described above. We expect some form of iterative retraining to be beneficial in active learning, but we currently have no concrete evidence to support this claim. We consider this a compelling direction for future research.

## 5 OOD GENERALIZATION

To get some measure of out-of-distribution generalization, we evaluate our large-scale ORM and PRM on a held-out set of 224 STEM questions, pulled from the most recent AP Physics, AP Calculus, AP Chemistry, AMC10, and AMC12 exams. Since these tests were released after the pre-training dataset was compiled, we can have high confidence that the model has not seen these problems. We report the best-of-100 performance of the ORM, PRM and majority voting in Table 1. We observe results similar to those in Section 3: the PRM outperforms both the ORM and majority voting. This shows us that the PRM can tolerate a modest amount of distribution shift and that its strong performance holds up on fresh test questions.

## 6 DISCUSSION

### 6.1 CREDIT ASSIGNMENT

One clear advantage of process supervision is that it provides more precise feedback than outcome supervision. A reward model trained with outcome supervision faces a difficult credit-assignment

task — to generalize well, it must determine where an incorrect solution went wrong. This is particularly difficult for hard problems: most model-generated solutions contain an error somewhere, so the marginal value of a negative label from outcome supervision is low. In contrast, process supervision provides a richer signal: it specifies both how many of the first steps were in fact correct, as well as the precise location of the incorrect step. Process supervision makes credit assignment easier, and we believe that this explains its strong performance.

## 6.2 ALIGNMENT IMPACT

Process supervision has several advantages over outcome supervision related to AI alignment. Process supervision is more likely to produce interpretable reasoning, since it encourages models to follow a process endorsed by humans. Process supervision is also inherently safer: it directly rewards an aligned chain-of-thought rather than relying on outcomes as a proxy for aligned behavior (Stuhlmüller & Byun, 2022). In contrast, outcome supervision is harder to scrutinize, and the preferences conveyed are less precise. In the worst case, the use of outcomes as an imperfect proxy could lead to models that become misaligned after learning to exploit the reward signal (Uesato et al., 2022; Cotra, 2022; Everitt et al., 2017).

In some cases, safer methods for AI systems can lead to reduced performance (Ouyang et al., 2022; Askell et al., 2021), a cost which is known as an alignment tax. In general, any alignment tax may hinder the adoption of alignment methods, due to pressure to deploy the most capable model. Our results show that process supervision in fact incurs a negative alignment tax. This could lead to increased adoption of process supervision, which we believe would have positive alignment side-effects. It is unknown how broadly these results will generalize beyond the domain of math, and we consider it important for future work to explore the impact of process supervision in other domains.

## 6.3 TEST SET CONTAMINATION

The test set of the MATH dataset contains problems that are discussed in several online venues, and it is likely that some of these problems appear in the pretraining dataset for our models. We attempted to remove all MATH problems from our MathMix dataset using string-matching heuristics, but since humans can post hard-to-detect rephrasings of a problem online, it is difficult to make any strong guarantees about the overlap between MathMix and the MATH dataset.

In our experience inspecting model-generated solutions, we saw no clear signs of our models memorizing MATH problems. However, it is impossible to rule out subtle forms of memorization that would slip past manual inspection, and it is still possible that some degree of contamination has slightly inflated our performance on the MATH test set. Even in that case, we would expect any contamination to manifest similarly across all methods, and that the relative comparisons made throughout this work would remain mostly unaffected.

We also note that the PRM regularly surfaces correct solutions to MATH problems that have a low single-digit percentage solve-rate under the generator, some examples of which can be seen in Appendix I. The generator's low solve-rate is an additional indication that it has not encountered such problems via test set contamination. Our generalization results from Section 5 further strengthen our claim that test set contamination has not significantly impacted this work, since we observe qualitatively similar results on problems that are guaranteed to be uncontaminated.

## 7 RELATED WORK

### 7.1 OUTCOME VS PROCESS SUPERVISION

In work closely related to our own, Uesato et al. (2022) compare the impact of outcome and process supervision in the domain of grade school math. They found that both methods led to similar final-answer error rates, and that process supervision achieved those results with less data. While our core methodology is very similar, there are three main details that differ. First, we use a more capable model to collect PRM800K dataset and to perform our large-scale experiments. However, our small-scale results in Section 4 suggest that large-scale models are not necessary to observe benefits from process supervision. Second, we evaluate on the MATH dataset, which is significantly

more challenging than GSM8K. Third, we collect a much larger quantity of process supervision data.

On the surface, the results from Uesato et al. (2022) may seem to conflict with our claim that process supervision leads to better performance. However, we believe the apparent conflict can be explained by the difference in the scale of the supervision. The data scaling trend in Figure 4a suggests that a small amount of process supervision and a large amount of outcome supervision do in fact lead to similar performance, consistent with the results from Uesato et al. (2022). The trend also shows that process supervision beats outcome supervision when scaled up, even when judged based solely on outcomes. This is consistent with our results in Section 3. We believe these results make a strong case for using process supervision.

### 7.2 SYNTHETIC SUPERVISION

Similar to our work in Section 4, Gao et al. (2022) use a large reward model to supervise the training of smaller models. They study the over-optimization that occurs during RLHF, with experiments that require large quantities of human preference data. To work around this challenge, they use a gold-standard reward model to replace human feedback. Our use of a large-scale reward model to supervise smaller reward models shares similarities with their approach.

### 7.3 NATURAL LANGUAGE REASONING

Several recent studies that have examined the reasoning ability of large language models are implicitly relevant to our work. Lewkowycz et al. (2022) showed that finetuning models on a large corpus of technical content led to significantly improved performance on MATH. Wang et al. (2022) showed that *self-consistency* leads to remarkably strong performance on many reasoning benchmarks, notably without requiring any additional finetuning. Wei et al. (2022) and Nye et al. (2021) demonstrate the importance of explicitly performing intermediate reasoning steps via a *chain of thought* or a *scratchpad* in order to solve tasks that require multi-step reasoning. Kojima et al. (2022) show that models are able to perform this behavior zero-shot, conditioned only on a simple prompt.

## 8 CONCLUSION

We have shown that process supervision can be used to train much more reliable reward models than outcome supervision in the domain of mathematical reasoning. We have also shown that active learning can be used to lower the cost of human data collection by surfacing only the most valuable model completions for human feedback. We release PRM800K, the full dataset of human feedback used to train our state-of-the-art reward model, with the hope that removing this significant barrier to entry will catalyze related research on the alignment of large language models. We believe that process supervision is currently under-explored, and we are excited for future work to more deeply investigate the extent to which these methods generalize.

#### ACKNOWLEDGMENTS

We thank Joshua Achiam, Mark Chen, Jonathan Gordon, Dan Hendrycks, Lukasz Kaiser, Oleg Murk, Ben Sokolowsky, Francis Song, and Jonathan Uesato for valuable feedback and thoughtful discussions; Giambattista Parascandolo and Daniel Selsam for their contributions to the MathMix dataset; Jonathan Ward for contributing to the data collection interface; Wojciech Zaremba for encouraging us to scale up data collection; Peter Hoeschele and Aris Kostantinidis for supporting our data collection; the research acceleration and supercomputing teams at OpenAI for providing infrastructure support; and the team at Scale and the many data-labelers who created PRM800K.

#### REPRODUCIBILITY STATEMENT

To assist reproducibility and further research we are releasing all of the labels that we gathered over the course of this project. Appendix B contains information about the dataset, what data was used for training, and a link to the repository containing the raw labels. Appendix C explains our evaluation methods. Finally, Appendix E and Appendix F explain how we trained our ORMs and PRMs.

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

# A  MATHMIX

Similar to Lewkowycz et al. (2022) we construct a large-scale dataset of high-quality math-relevant tokens for use in a lightweight pretraining stage, before finetuning on comparably smaller datasets like MATH and PRM800K. This dataset, which we call MathMix, has two main differences compared to the one used to train Minerva. First, it is smaller and more aggressively filtered to high-quality math problem-solving content, and second, it does not explicitly mix in general language data.

Minerva was trained on 38.5B tokens of arXiv documents and webscrape pages with LaTeX content, while MathMix consists of a smaller set of 1.5B tokens containing individual math problems and their solutions, free-form text discussing math problems and concepts, and synthetic data (Table 2). While Minerva was pretrained on a dataset with 5% general natural language data, we chose not to mix in any natural language data explicitly, primarily because MathMix already contains plenty of natural language data.

Table 2: MathMix dataset components.

| Data type | Token count | Present in pretraining? |
|---|---|---|
| Math problems and solutions | $\sim 275M$ | No |
| Free-form math discussion text (1) | $\sim 430M$ | No |
| Free-form math discussion text (2) | $\sim 450M$ | Yes |
| Synthetic data (1) | $\sim 30M$ | No |
| Synthetic data (2) | $\sim 100M$ | Yes |
| Critiques grading data | $\sim 500M$ | No |

Note that when training smaller models, as in Section 4, we use a slightly smaller variant of MathMix that excludes the critiques data and only consists of 1B tokens. For our large models experiments, we train on MathMix for roughly 3B tokens (2 epochs). For our small models experiments, we train for 6 epochs (roughly 6.6B tokens).

We apply a set of decontamination checks on MathMix against the test split of the MATH dataset, including stripping out LaTeX and searching for matching n-grams, but we can make no strong guarantees on the efficacy of this decontamination. As discussed in Section 6.3, we would not expect the relative comparisons made throughout this work to be significantly impacted by test set contamination.

## B  PRM800K

We collected 1,085,590 step-level labels over 101,599 solution samples. We present the whole unfiltered dataset as PRM800K. During training we discard labels used for quality control, as well as any step-level labels for which the labeler was unable to complete the task. The filtered dataset contains about 800,000 step-level labels over 75,000 solutions. The full PRM800K dataset is available at https://github.com/openai/prm800k.

The data collection was split into two separate phases. In phase 1, we collected labels for multiple alternative completions at each step of a solution. This seeded our dataset but was cumbersome— for many steps the alternatives were repetitive, and we found labelers spent a lot of time supervising long uninteresting solutions. As a result, the step-level labels we collected in this phase are more repetitive than those collected later. In total, phase 1 represents about 5% of PRM800K, or about 40,000 step-level labels.

The majority of our labels were collected as part of phase 2, during which we scaled up and streamlined the data collection process. Phase 2 data collection is split into 10 generations. For each generation, we sample $N$ solutions per problem from the generator. We rank these solutions with our current best PRM and surface the highest scoring wrong-answer solutions to our labelers. We retrain this PRM between each generation using all the latest data. This active learning strategy changes the balance of our data considerably. Though we sometimes surfaced correct solutions (either by manually injecting correct solutions or because of errors in our automatic grading), the vast majority of the labels we collected in this phase are for incorrect solutions. Table 3 breaks down the balance of correct/incorrect steps and solutions between the different phases of data collection. Though we mostly collected labels on incorrect solutions, we still collected many labels for correct individual steps. In fact, our small-scale ablations in Section 4.2 suggest that this active learning strategy, which favors labelling high-scoring wrong-answer solutions, improves performance despite the resulting imbalance in the dataset.

Table 3: Distribution of positive/negative steps/solutions.

|                          | phase 1 | phase 2 | combined |
|--------------------------|---------|---------|----------|
| % end in correct solution | 85.1    | 13.2    | 14.2     |
| % correct steps           | 58.6    | 74.1    | 73.1     |

Some of our phase 2 questions are intended for quality control. For a quality control question, researchers mark which steps are reasonable to label as incorrect. Then we assess that labelers are able to consistently mark those steps as incorrect. Prior to starting on phase 2, we required all labelers to label 30 quality control questions. This served as a screening test, and we only admitted labelers that agreed with our gold labels at least 75% of the time.

We then designated 10-20 problems per generation as additional quality control questions, and we randomly served them to labelers as they worked through the task. We used the results of this continuous quality control to remove labelers whose quality slipped too far, as well as to prepare educational material on common mistakes in order to improve labeler alignment with our instructions.

## C  EVALUATION

As we scaled up the project, we began having to collect labels on multiple solutions for the same training problem. In order to avoid the risk of over-fitting on the 7,500 MATH training problems, we expanded the training set to include 4,500 MATH test split problems. We therefore evaluate our models only on the remaining 500 held-out problems. We selected these 500 test problems uniformly at random. In Figure 5, we show that the distribution of difficulty levels and subjects in this subset is representative of the MATH test set as a whole. The specific test set we used can be found at https://github.com/openai/prm800k. We leave it for future work to explore how many distinct training problems are actually necessary, and how quickly our methods overfit to the training set.

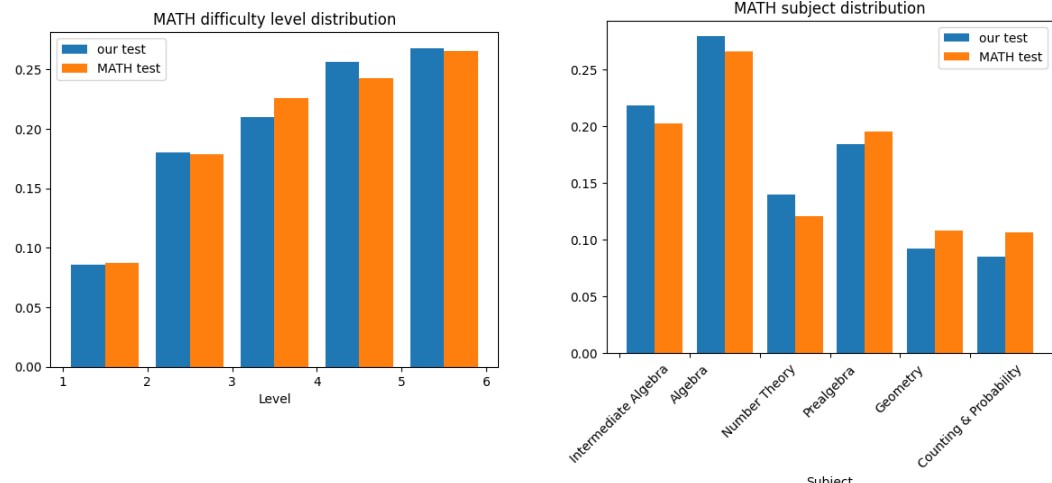

Figure 5: Two histograms comparing the distribution of problem difficulty levels and subjects in both the original MATH test set and in our 500 problem test subset.

## D  LABELLING INSTRUCTIONS

Labelers were tasked to look at steps in a solution and label each one as **positive**, **negative**, or **neutral**. A step is considered **neutral** if it is appropriate in context, reasonable, correct, and contains only computations that can be verified easily. A step is **positive** if it is **neutral** and also progresses towards the solution. All other steps are considered **negative**. Labelers were not given reference solutions, but they were given the ground truth final answers. We chose not to provide reference solutions to avoid biasing them towards one particular path to the solution. We chose to provide ground truth final answers since this information can sometimes help labelers resolve their own misunderstandings.

In phase 1, labelers were permitted to enter their own steps in the case that all candidate steps were **negative**. Then the solution would progress from a randomly selected **positive** step (or **neutral** if their were no **positive** ones). This often resulted in trajectories that got stuck in endless sequences of **neutral** steps that said reasonable things but made frustratingly slow progress towards a solution or **negative** steps that needed constant human supervision. In phase 2, we pre-generate whole solutions and end the task as soon as the first **negative** step is encountered. The full instructions given to labelers can be found at https://github.com/openai/prm800k/tree/main/prm800k/instructions.

## E  ORM TRAINING DETAILS

We train outcome-supervised reward models in the same manner as token-level verifiers from Cobbe et al. (2021), with a few subtle differences to hyperparameters. In particular, we only train for a single epoch on each dataset of model samples and reward model labels, without dropout, and without jointly learning a language modeling objective. We find that performance is not sensitive to most other hyperparameters, within a reasonable range.

To collect model samples, we simply sample uniformly from the generator at a temperature of 1.0 without applying any rebalancing of positives or negatives. At training time, the reward model makes predictions for every token in the context. The target for each token in a solution is the same, based on whether the solution is labelled correct or incorrect. At test time, we simply use the score of the final token in the completion as the overall score of the solution. We note that this setup is identical to the way token-level verifiers were trained in Cobbe et al. (2021).

# F   PRM Details

## F.1   Training

We train our PRMs by fine-tuning the MathMix model to predict the probability of positive, negative, and neutral labels given a solution prefix ending in one of our labeled steps. We sweep over hyperparameters using a dataset containing the first $\sim 10\%$ of PRM800K. Fine-tuning an LLM from its ordinary language modeling task to a classification task like this is a large distribution shift, and we found low learning rates were important to stable PRM training.

All of our PRMs are trained for 2 epochs. On smaller datasets (such as in phase 1 and the first few generations of phase 2) this improves the final performance over training for just 1 epoch. Additional epochs, up to some point, don't noticeably help or hurt performance. On larger datasets, the benefits of 2 epoch training diminishes, but we continue doing it for consistency.

## F.2   Scoring

There are multiple ways of using the PRM to score solutions. In general, we produce a single solution-level score by performing a reduction over step-level scores, where the step-level score is the probability that the step's label is positive. This involves two specific implementation decisions. First, when determining a step-level score, we either consider a neutral label to be positive or negative. Second, when determining a solution-level score, we either use the minimum or the product over step-level scores as a reduction.

We show results from all four scoring strategies in Table 4. The best performing strategy is to take the product of step-level scores and to consider the neutrals as positives, but the difference in performance between all strategies is minor. Throughout the rest of this work, we consider neutral steps to be positive, and we define the solution score to be the product of step-level scores. Using the product instead of the minimum as the reduction does create a slight bias against solutions with a larger number of steps.

Table 4: Best-of-1860 test performance using the PRM with four different scoring strategies.

|                    | product | minimum |
|--------------------|---------|---------|
| neutral = positive | 78.2%   | 77.6%   |
| neutral = negative | 77.4%   | 77.8%   |

# G    DIFFICULTY BREAKDOWN

We show performance of our ORM and PRM on each quintile of the MATH dataset. We determine quintiles based on the pass rate under the generator. It is interesting to note that the performance gap is not only apparent on high difficulty problems: it is in fact apparent across all difficulties. For the lowest difficulty problems, we see that it is possible to find adversarial examples that fool the ORM, since the ORM's performance slightly decreases as the number of samples increases. In contrast, the PRM remains highly robust over this same set of samples.

We also see that increasing the number of samples has the largest positive effect on the highest difficulty problems. This is to be expected, since a large number of generator samples may be required to find a true and convincing solution to a hard problem.

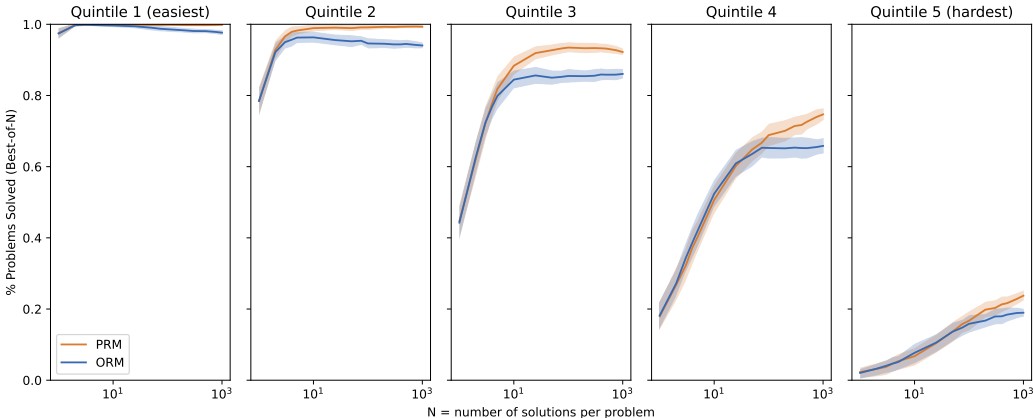

Figure 6: A breakdown of ORM vs PRM performance by problem difficulty.

## H  SYNTHETIC SUPERVISION DETAILS

We can use $\text{PRM}_{\text{large}}$ to provide either outcome or process supervision for smaller models. We determine the labels for individual steps based on the step-level probabilities outputted by $\text{PRM}_{\text{large}}$. To do this, we set an arbitrary threshold: any step that $\text{PRM}_{\text{large}}$ assigns a negative label with greater than 20% probability is considered incorrect. We choose this threshold based on the observation that $\text{PRM}_{\text{large}}$ is slightly miscalibrated in the direction of favoring positive labels.

To provide process supervision for a solution, we directly return the step-level labels (positive or negative) provided by $\text{PRM}_{\text{large}}$, up until the first step that is marked as negative. This mimics our true human data collection process. To provide outcome supervision, we mark the solution as correct if and only if $\text{PRM}_{\text{large}}$ considers every step to be correct (using the same thresholding logic).

# I   PRM VISUALIZATIONS

All examples shown come from the large-scale generator (GPT-4). We note the pass-rate under the generator to give some sense of the difficulty of these problems.

## I.1   TRUE POSITIVES

These cherry-picked examples show the best-of-1860 solution from the generator as ranked by the large-scale PRM.

Problem 1. Generator pass-rate: $0.1\%$. This challenging trigonometry problem requires applying several identities in a not-at-all obvious succession. Most solution attempts fail, because it is hard to choose which identities are actually helpful. Though successful solutions to this problem are rare, the reward model correctly recognizes when a valid chain-of-thought has been found.

**Simplify** $\tan 100° + 4\sin 100°$.

I notice that the angle 100 degrees is not a special angle, so I will need to use some trigonometric identities to simplify this expression.

I recall that $\tan\theta = \frac{\sin\theta}{\cos\theta}$ and $\sin 2\theta = 2\sin\theta\cos\theta$.

I can use these identities to rewrite the expression as $\frac{\sin 100°}{\cos 100°} + 4\sin 100°$.

I can combine the terms by finding a common denominator of $\cos 100°$.

I get $\frac{\sin 100° + 4\sin 100°\cos 100°}{\cos 100°}$.

I can simplify the numerator by using the double angle identity again, but in reverse.

I get $\frac{\sin 100° + 2\sin 200°}{\cos 100°}$.

I notice that 200 degrees is the same as 180 degrees plus 20 degrees, so I can use the sum angle identity for sine.

I get $\frac{\sin 100° + 2(\sin 180°\cos 20° + \cos 180°\sin 20°)}{\cos 100°}$.

I recall that $\sin 180° = 0$ and $\cos 180° = -1$, so I can simplify further.

I get $\frac{\sin 100° - 2\sin 20°}{\cos 100°}$.

I notice that 100 degrees is the same as 90 degrees plus 10 degrees, so I can use the sum angle identity for sine and cosine.

I get $\frac{(\sin 90°\cos 10° + \cos 90°\sin 10°) - 2\sin 20°}{(\cos 90°\cos 10° - \sin 90°\sin 10°)}$.

I recall that $\sin 90° = 1$ and $\cos 90° = 0$, so I can simplify further.

I get $\frac{\cos 10° - 2\sin 20°}{-\sin 10°}$.

I can simplify the sign by multiplying the numerator and denominator by -1.

I get $\frac{2\sin 20° - \cos 10°}{\sin 10°}$.

I notice that 20 degrees is the same as 30 degrees minus 10 degrees, so I can use the difference angle identity for sine.

I get $\frac{2(\sin 30°\cos 10° - \cos 30°\sin 10°) - \cos 10°}{\sin 10°}$.

I recall that $\sin 30° = \frac{1}{2}$ and $\cos 30° = \frac{\sqrt{3}}{2}$, so I can simplify further.

I get $\frac{\cos 10° - \sqrt{3}\sin 10° - \cos 10°}{\sin 10°}$.

I can cancel out the $\cos 10°$ terms in the numerator.

I get $\frac{-\sqrt{3}\sin 10°}{\sin 10°}$.

I can cancel out the $\sin 10°$ terms in the numerator and denominator.

I get $-\sqrt{3}$.

Answer: $-\sqrt{3}$

**Problem 2.** Generator pass-rate: $5.8\%$. In step 7 and 8, the generator starts performing guess-and-check. This is a common place the model might hallucinate, by claiming a particular guess is successful when it isn't. In this case, the reward model verifies each step and determines that the chain-of-thought is correct.

> **Find the unique odd integer $t$ such that $0 < t < 23$ and $t + 2$ is the inverse of $t$ modulo 23.**
>
> I need to find an odd integer $t$ that satisfies the congruence $t + 2 \equiv t^{-1} \pmod{23}$.
>
> To do that, I can multiply both sides by $t$ and get $t^2 + 2t \equiv 1 \pmod{23}$.
>
> Then I can complete the square by adding and subtracting $(2/2)^2 = 1$ to the left side and get $(t+1)^2 - 1 \equiv 1 \pmod{23}$.
>
> Simplifying, I get $(t+1)^2 \equiv 2 \pmod{23}$.
>
> Now I need to find a square root of 2 modulo 23.
>
> I can try some values of $t + 1$ and see if they work.
>
> For example, if $t + 1 = 3$, then $(t+1)^2 = 9 \not\equiv 2 \pmod{23}$.
>
> If $t + 1 = 5$, then $(t+1)^2 = 25 \equiv 2 \pmod{23}$.
>
> So one possible value of $t$ is $t = 5 - 1 = 4$.
>
> But this is not odd, so I need to find another one.
>
> I know that if $t + 1$ is a square root of 2 modulo 23, then so is $-t - 1$.
>
> So I can try $-t - 1 = 5$ and get $t = -6$.
>
> But this is negative, so I need to add 23 to get a positive value.
>
> So another possible value of $t$ is $t = -6 + 23 = 17$.
>
> This is odd and satisfies the original congruence, so this is the unique solution.
>
> Answer: 17

**Problem 3.** Generator pass-rate: $1.7\%$. The generator successfully applies several trigonometric identities to simplify the expression.

> **Find the degree measure of the least positive angle $\theta$ for which**
> $$\tan\theta = \frac{\cos 5° \cos 20° + \cos 35° \cos 50° - \sin 5° \sin 20° - \sin 35° \sin 50°}{\sin 5° \cos 20° - \sin 35° \cos 50° + \cos 5° \sin 20° - \cos 35° \sin 50°}.$$
>
> I notice that the numerator and denominator of the fraction are both sums of products of cosine and sine functions of angles that add up to 25 degrees or 85 degrees.
>
> This suggests that I might be able to use some trigonometric identities to simplify the expression.
>
> For example, I recall that $\cos(A + B) = \cos A \cos B - \sin A \sin B$ and $\sin(A + B) = \sin A \cos B + \cos A \sin B$.
>
> If I apply these identities to the terms in the numerator, I get $\cos 5° \cos 20° + \cos 35° \cos 50° - \sin 5° \sin 20° - \sin 35° \sin 50° = \cos(5° + 20°) + \cos(35° + 50°) = \cos 25° + \cos 85°$.
>
> Similarly, if I apply these identities to the terms in the denominator, I get $\sin 5° \cos 20° - \sin 35° \cos 50° + \cos 5° \sin 20° - \cos 35° \sin 50° = \sin(5° + 20°) - \sin(35° + 50°) = \sin 25° - \sin 85°$.
>
> Now I have $\tan\theta = \frac{\cos 25° + \cos 85°}{\sin 25° - \sin 85°}$.
>
> I wonder if I can simplify this fraction further.
>
> I recall another trigonometric identity that says $\cos A + \cos B = 2\cos\frac{A+B}{2}\cos\frac{A-B}{2}$.
>
> If I apply this identity to the numerator, I get $\cos 25° + \cos 85° = 2\cos\frac{25°+85°}{2}\cos\frac{25°-85°}{2} = 2\cos 55° \cos(-30°)$.
>
> I also recall that $\cos(-\theta) = \cos\theta$, so I can write $\cos(-30°) = \cos 30°$.
>
> Similarly, I recall another trigonometric identity that says $\sin A - \sin B = 2\cos\frac{A+B}{2}\sin\frac{A-B}{2}$.
>
> If I apply this identity to the denominator, I get $\sin 25° - \sin 85° = 2\cos\frac{25°+85°}{2}\sin\frac{25°-85°}{2} = 2\cos 55° \sin(-30°)$.
>
> I also recall that $\sin(-\theta) = -\sin\theta$, so I can write $\sin(-30°) = -\sin 30°$.
>
> Now I have $\tan\theta = \frac{2\cos 55° \cos 30°}{2\cos 55°(-\sin 30°)} = -\frac{\cos 30°}{\sin 30°}$.
>
> I know that $\cos 30° = \frac{\sqrt{3}}{2}$ and $\sin 30° = \frac{1}{2}$, so I can simplify this to $\tan\theta = -\frac{\sqrt{3}}{1} = -\sqrt{3}$.
>
> I also know that $\tan\theta$ is negative in the second and fourth quadrants, and that $\tan\theta = -\sqrt{3}$ when $\theta$ is 120° or 300°.
>
> Therefore, the least positive angle $\theta$ for which $\tan\theta = -\sqrt{3}$ is 120°.
>
> Answer: 120

Problem 4. Generator pass-rate: $4.5\%$. Here, the generator successfully performs a complex series of polynomial factorizations. The use of the Sophie-Germain identity in step 5 is an important step that could be considered insightful.

**Let**
$$x^8 + 3x^4 - 4 = p_1(x)p_2(x)\cdots p_k(x),$$
where each non-constant polynomial $p_i(x)$ is monic with integer coefficients, and cannot be factored further over the integers. Compute $p_1(1) + p_2(1) + \cdots + p_k(1)$.

I notice that the given polynomial has even degree and only even powers of $x$, so I can try to make a substitution to simplify it.

Let $y = x^4$, then the polynomial becomes $y^2 + 3y - 4$, which is a quadratic equation.

I can factor this quadratic equation as $(y+4)(y-1)$, so the original polynomial is $(x^4+4)(x^4-1)$.

Now I need to factor each of these factors further, if possible, over the integers.

For the first factor, $x^4 + 4$, I recall the Sophie Germain identity, which says that $a^4 + 4b^4 = (a^2 + 2b^2 + 2ab)(a^2 + 2b^2 - 2ab)$.

Applying this identity with $a = x$ and $b = 1$, I get $x^4 + 4 = (x^2 + 2x + 2)(x^2 - 2x + 2)$.

For the second factor, $x^4 - 1$, I can use the difference of squares formula, which says that $a^2 - b^2 = (a+b)(a-b)$.

Applying this formula with $a = x^2$ and $b = 1$, I get $x^4 - 1 = (x^2 + 1)(x^2 - 1)$.

Then I can apply the difference of squares formula again with $a = x$ and $b = 1$, to get $x^2 - 1 = (x+1)(x-1)$.

So the original polynomial is $(x^4+4)(x^4-1) = (x^2 + 2x + 2)(x^2 - 2x + 2)(x^2 + 1)(x+1)(x-1)$.

These are all the irreducible factors over the integers, so $p_1(x) = x^2 + 2x + 2$, $p_2(x) = x^2 - 2x + 2$, $p_3(x) = x^2 + 1$, $p_4(x) = x + 1$, and $p_5(x) = x - 1$.

To find the sum of their values at $x = 1$, I just need to plug in $x = 1$ into each factor and add them up.

I get $p_1(1) = 1^2 + 2 \cdot 1 + 2 = 5$, $p_2(1) = 1^2 - 2 \cdot 1 + 2 = 1$, $p_3(1) = 1^2 + 1 = 2$, $p_4(1) = 1 + 1 = 2$, and $p_5(1) = 1 - 1 = 0$.

So the final answer is $5 + 1 + 2 + 2 + 0 = 10$.

Answer: 10

## I.2 TRUE NEGATIVES

Problem 5. Generator pass-rate: $4.5\%$. The generator attempts to use the difference of squares formula in step 12 on an expression that isn't in fact a difference of squares. The reward model catches this mistake.

**Let**
$$x^8 + 3x^4 - 4 = p_1(x)p_2(x)\cdots p_k(x),$$
where each non-constant polynomial $p_i(x)$ is monic with integer coefficients, and cannot be factored further over the integers. Compute $p_1(1) + p_2(1) + \cdots + p_k(1)$.

This problem is about factoring a polynomial over the integers, which can be a tricky task.

One possible strategy is to look for rational roots of the polynomial, using the rational root theorem.

However, that would involve checking a lot of possible factors of $-4$, and there is no guarantee that the polynomial has any rational roots at all.

Another possible strategy is to use some algebraic tricks to simplify the polynomial or reduce its degree.

I notice that the polynomial has only even powers of $x$, so I can try to substitute $y = x^2$ and see what happens.

Then the polynomial becomes $y^4 + 3y^2 - 4$, which is a quadratic equation in $y^2$.

I can try to factor this quadratic equation by looking for two numbers that multiply to $-4$ and add to 3.

Those numbers are 4 and $-1$, so I can write $y^4 + 3y^2 - 4 = (y^2 + 4)(y^2 - 1)$.

Now I can substitute back $x^2 = y$ and get $x^8 + 3x^4 - 4 = (x^4 + 4)(x^4 - 1)$.

This looks promising, because I have reduced the degree of the factors from 8 to 4.

I can try to factor each of these factors further, using the difference of squares formula.

I get $x^8 + 3x^4 - 4 = (x^2 + 2)(x^2 - 2)(x^2 + 1)(x^2 - 1)$.

I can apply the difference of squares formula again to the last factor and get $x^8 + 3x^4 - 4 = (x^2 + 2)(x^2 - 2)(x^2 + 1)(x+1)(x-1)$.

Now I have factored the polynomial completely into monic linear and quadratic factors with integer coefficients.

These are the $p_i(x)$'s that the problem is asking for.

To find the sum of their values at $x = 1$, I just need to plug in $x = 1$ into each factor and add them up.

I get $p_1(1) + p_2(1) + \cdots + p_k(1) = (1^2 + 2)(1^2 - 2)(1^2 + 1)(1+1)(1-1)$.

Simplifying, I get $p_1(1) + p_2(1) + \cdots + p_k(1) = (3)(-1)(2)(2)(0)$.

Multiplying, I get $p_1(1) + p_2(1) + \cdots + p_k(1) = 0$.

Answer: 0

**Problem 6.** Generator pass-rate: $93.5\%$. In step 7, the generator makes an incorrect attempt to simplify an expression. The reward model catches this mistake.

**A class of 30 students recently took a test. If 20 students scored 80, 8 students scored 90, and 2 students scored 100, then what was the class average (the mean) on this test?**

To find the class average, I need to add up all the scores and divide by the number of students.

I can use a shortcut to add up the scores, since they are grouped by value.

For example, instead of adding $80 + 80 + ... + 80$ twenty times, I can just multiply 80 by 20 to get the same result.

Similarly, I can multiply 90 by 8 and 100 by 2 to get the sums of those groups.

So, the total sum of the scores is 80 * 20 + 90 * 8 + 100 * 2.

I can use the distributive property of multiplication to simplify this expression.

It is equivalent to 80 * (20 + 1) + 90 * 8 + 100.

That is, 80 * 21 + 720 + 100.

I can do the arithmetic in my head or use a calculator.

80 * 21 is 1680, so the total sum is $1680 + 720 + 100$, which is 2500.

Now, to find the average, I need to divide 2500 by 30, the number of students.

I can use long division or a calculator for this step.

2500 divided by 30 is 83.33, repeating.

This means that the class average is 83.33, or 83 and one-third.

Answer: 83.33

**Problem 7.** Generator pass-rate: $48.0\%$. In step 11, the generator makes a simple calculation error. The reward model catches this mistake.

**If $A$ is the sum of the positive divisors of $500$, what is the sum of the distinct prime divisors of $A$?**

I notice that $500 = 2^2 \cdot 5^3$, so its positive divisors are of the form $2^a \cdot 5^b$, where $a \in \{0, 1, 2\}$ and $b \in \{0, 1, 2, 3\}$.

To find the sum of these divisors, I can use a formula that involves the geometric series: if $p$ is a prime number and $n$ is a nonnegative integer, then the sum of the powers of $p$ from 0 to $n$ is $\frac{p^{n+1}-1}{p-1}$.

So, the sum of the powers of 2 from 0 to 2 is $\frac{2^3-1}{2-1} = 7$, and the sum of the powers of 5 from 0 to 3 is $\frac{5^4-1}{5-1} = 156$.

By the distributive property, the sum of the divisors of 500 is the product of these two sums: $A = 7 \cdot 156 = 1092$.

Now, I need to find the sum of the distinct prime divisors of $A$.

I can try to factor $A$ into primes by trial division.

I notice that $A$ is divisible by 2, so I divide by 2 and get 546.

I repeat this process and divide by 2 again, getting 273.

Now, 273 is not divisible by 2, so I try the next prime, 3.

I see that 273 is divisible by 3, so I divide by 3 and get 91.

I divide by 3 again, getting 31.

Now, 31 is a prime number, so I have found all the prime factors of $A$: $A = 2^2 \cdot 3^2 \cdot 31$.

The sum of the distinct prime divisors of $A$ is $2 + 3 + 31 = 36$.

Answer: 36

**Problem 8.** Generator pass-rate: $5.8\%$. The justification in step 8 is strange, but the reward model lets it slide. In step 9, though, the model incorrectly factors the expression. The reward model catches this mistake.

> **Find the unique odd integer $t$ such that $0 < t < 23$ and $t + 2$ is the inverse of $t$ modulo $23$.**
>
> I need to find an odd integer $t$ between 0 and 23 such that $t + 2$ is the inverse of $t$ modulo 23.
>
> That means that $t$ and $t + 2$ multiply to give 1 modulo 23.
>
> I can write this as $t(t + 2) \equiv 1 \pmod{23}$.
>
> I can try to guess and check values of $t$ until I find one that works, but that might take a long time.
>
> I can also try to use some properties of modular arithmetic to simplify the equation.
>
> For example, I can subtract 23 from $t + 2$ and still get the same remainder modulo 23.
>
> So I can write $t(t + 2) \equiv t(t - 21) \pmod{23}$.
>
> This looks simpler, because now I have a difference of two multiples of $t$.
>
> I can factor out $t$ and get $t(t - 21) \equiv t(1 - 21) \pmod{23}$.
>
> Now I can divide both sides by $t$, as long as $t$ is not a multiple of 23.
>
> This gives me $t - 21 \equiv 1 - 21 \pmod{23}$.
>
> I can simplify this further by adding 21 to both sides.
>
> This gives me $t \equiv 1 \pmod{23}$.
>
> This means that $t$ has the same remainder as 1 when divided by 23.
>
> The smallest positive value of $t$ that satisfies this is $t = 1$.
>
> This is also an odd integer, so it meets the criteria of the problem.
>
> I can check that $t = 1$ works by plugging it into the original equation.
>
> I get $1(1 + 2) \equiv 1 \pmod{23}$, which is true.
>
> Therefore, the unique odd integer $t$ that satisfies the given conditions is $t = 1$.
>
> Answer: 1

### I.3 FALSE POSITIVES

**Problem 9.** Generator pass-rate: $18.5\%$. The generator makes a subtle counting error in step 9. On the surface, it appears reasonable to claim that there are 5 ways to exchange the same colored ball since there are 5 colors. However, this undercounts by a factor of 2, since Bob has 2 choices for which ball to return to Alice. The reward model is fooled by this mistake.

> **Bob and Alice each have a bag that contains one ball of each of the colors, blue, green, orange, red, and violet. Alice randomly selects one ball from her bag and puts it into Bob's bag. Bob then randomly selects one ball from his bag and puts it into Alice's bag. What is the probability that after this process the contents of the two bags are the same?**
>
> To find the probability of an event, I need to divide the number of favorable outcomes by the number of possible outcomes.
>
> In this case, the event is that the contents of the two bags are the same after the process.
>
> So I need to count how many ways this can happen, and how many ways the process can happen in total.
>
> I will start with the total number of ways the process can happen.
>
> Alice can choose any of the five balls in her bag, and Bob can choose any of the six balls in his bag.
>
> So the total number of ways the process can happen is 5 times 6, which is 30.
>
> Now I will count how many ways the contents of the two bags can be the same after the process.
>
> This means that Alice and Bob must have exchanged the same color of ball.
>
> There are five colors, so there are five ways they can exchange the same color of ball. **ERROR**
>
> Therefore, the number of favorable outcomes is 5.
>
> So the probability of the event is 5 divided by 30, which simplifies to 1/6.
>
> Answer: 1/6

Problem 10. Generator pass-rate: $17.6\%$. In step 13, the generator attempts to simplify the equation by combining like terms. It correctly moves and combines the linear terms to the left-hand side, but then mistakenly leaves the right-hand side untouched. The reward model is fooled by this mistake.

Denali and Nate work for a dog walking business and are paid for each dog they walk. Denali is responsible for $16$ dogs and Nate is responsible for $12$ dogs. Under the company's new policy, they will be assigned or unassigned new dogs in groups of $x$ dogs. The ratio of Denali's pay to Nate's pay would be the same if Denali started walking $4x$ more dogs and Nate stayed at $12$ dogs or if $x$ of Nate's dogs were reassigned to Denali. Find $x$ if $x \neq 0$.

I need to find a value of $x$ that makes the ratio of Denali's pay to Nate's pay the same in two different scenarios.

I can assume that the pay per dog is constant, so the ratio of their pay is the same as the ratio of the number of dogs they walk.

Let me write an equation for the first scenario, where Denali walks $4x$ more dogs and Nate stays at 12 dogs.

The ratio of Denali's dogs to Nate's dogs is $\frac{16+4x}{12}$.

Now let me write an equation for the second scenario, where $x$ of Nate's dogs are reassigned to Denali.

The ratio of Denali's dogs to Nate's dogs is $\frac{16+x}{12-x}$.

I want these two ratios to be equal, so I can set them equal to each other and solve for $x$.

I get $\frac{16+4x}{12} = \frac{16+x}{12-x}$.

I can cross-multiply and simplify this equation.

I get $(16 + 4x)(12 - x) = (16 + x)(12)$.

Expanding the brackets, I get $192 - 16x + 48x - 4x^2 = 192 + 12x$.

Subtracting 192 from both sides, I get $-16x + 48x - 4x^2 = 12x$.

Rearranging the terms, I get $-4x^2 + 20x = 12x$. **ERROR**

Subtracting $12x$ from both sides, I get $-4x^2 + 8x = 0$.

I can factor out $-4x$ from both terms, and get $-4x(x - 2) = 0$.

This means that either $-4x = 0$ or $x - 2 = 0$.

Solving for $x$, I get $x = 0$ or $x = 2$.

However, the problem says that $x \neq 0$, so I can ignore that solution.

Therefore, the only valid solution is $x = 2$.

Answer: 2

Problem 11. Generator pass-rate: $13.4\%$. The generator attempts to perform long division, but in step 16, it forgets to include the leading zeros in the repeating part of the decimal. The reward model is fooled by this mistake.

When converting $\frac{31}{11111}$ to a decimal, the decimal turns out to be a repeating decimal. How many digits repeat in this repeating decimal? For example, if you get the repeating decimal $0.\overline{123}$, then your answer should be $3$, and if you get $0.436\overline{7}$, your answer should be $1$.

To convert a fraction to a decimal, I can divide the numerator by the denominator using long division.

I'll start by writing 31 over 11111 and bring down a zero.

Then I'll ask, how many times does 11111 go into 310?

Well, it doesn't go in evenly, so I'll write a zero above the decimal point and bring down another zero.

Now I have 3100.

How many times does 11111 go into 3100?

Still zero, so I'll write another zero above the decimal point and bring down another zero.

Now I have 31000.

How many times does 11111 go into 31000?

Ah, now I can see that it goes in twice, so I'll write a 2 above the decimal point and subtract 22222 from 31000 to get 8778.

I'll bring down another zero and ask, how many times does 11111 go into 87780?

It goes in seven times, so I'll write a 7 above the decimal point and subtract 77777 from 87780 to get 10003.

I'll bring down another zero and ask, how many times does 11111 go into 100030?

It goes in nine times, so I'll write a 9 above the decimal point and subtract 99999 from 100030 to get 31.

Wait a minute, I'm back to where I started! That means the decimal will repeat from here on.

So I'll put a bar over the 279 to indicate that it repeats indefinitely. **ERROR**

The decimal is 0.00279 with a bar over 279.

The question asks how many digits repeat in this repeating decimal.

I can see that there are three digits that repeat: 2, 7, and 9.

So the answer is 3.

Answer: 3

Problem 12. Generator pass-rate: $9.1\%$. In step 4, the generator falsely claims that the sequence repeats itself every 12 terms, when it's in fact every 10 terms. This sort of counting mistake occasionally fools the reward model.

**Given that**

$$x_1 = 211,$$
$$x_2 = 375,$$
$$x_3 = 420,$$
$$x_4 = 523, \text{ and}$$
$$x_n = x_{n-1} - x_{n-2} + x_{n-3} - x_{n-4} \text{ when } n \geq 5,$$

**find the value of** $x_{531} + x_{753} + x_{975}$.

This is a problem involving a recursive sequence, where each term depends on the previous four terms.

One way to approach this problem is to look for a pattern or a formula that can help us find any term of the sequence without having to compute all the previous terms.

To do that, let's write down the first few terms of the sequence and see if we notice anything:

$$x_1 = 211,$$
$$x_2 = 375,$$
$$x_3 = 420,$$
$$x_4 = 523,$$
$$x_5 = x_4 - x_3 + x_2 - x_1 = 523 - 420 + 375 - 211 = 267,$$
$$x_6 = x_5 - x_4 + x_3 - x_2 = 267 - 523 + 420 - 375 = -211,$$
$$x_7 = x_6 - x_5 + x_4 - x_3 = -211 - 267 + 523 - 420 = -375,$$
$$x_8 = x_7 - x_6 + x_5 - x_4 = -375 - (-211) + 267 - 523 = -420,$$
$$x_9 = x_8 - x_7 + x_6 - x_5 = -420 - (-375) + (-211) - 267 = -523,$$
$$x_{10} = x_9 - x_8 + x_7 - x_6 = -523 - (-420) + (-375) - (-211) = -267,$$
$$x_{11} = x_{10} - x_9 + x_8 - x_7 = -267 - (-523) + (-420) - (-375) = 211,$$
$$x_{12} = x_{11} - x_{10} + x_9 - x_8 = 211 - (-267) + (-523) - (-420) = 375,$$
$$x_{13} = x_{12} - x_{11} + x_{10} - x_9 = 375 - 211 + (-267) - (-523) = 420,$$
$$x_{14} = x_{13} - x_{12} + x_{11} - x_{10} = 420 - 375 + 211 - (-267) = 523.$$

We can see that the sequence repeats itself every 12 terms, starting from $x_1$. **ERROR**

This means that we can find any term of the sequence by finding its remainder when divided by 12, and then looking up the corresponding term in the first cycle.

For example, to find $x_{531}$, we can divide 531 by 12 and get a quotient of 44 and a remainder of 3.

This means that $x_{531}$ is the same as $x_3$, which is 420.

Similarly, we can find $x_{753}$ and $x_{975}$ by dividing them by 12 and looking up their remainders.

We get that $x_{753}$ has a remainder of 9, so it is the same as $x_9$, which is -523.

And $x_{975}$ has a remainder of 3, so it is the same as $x_3$, which is 420.

Therefore, the value of $x_{531} + x_{753} + x_{975}$ is $420 + (-523) + 420 = 317$.

Answer: 317

