# OpenReview forum: "Let's Verify Step by Step"
_ICLR.cc/2024/Conference — ICLR 2024 poster_

### Official Review · Reviewer_5YZq · 2023-10-28

**Soundness:** 4 excellent
**Presentation:** 3 good
**Contribution:** 4 excellent
**Rating:** 8
**Confidence:** 3

**Summary:**

In this work, the authors conducted an investigation to determine whether the outcome supervision method or the process supervision method is more effective for training reliable models. They found that process supervision produces more reliable models, and that a large reward model can accurately approximate human supervision for smaller reward models. Additionally, they discovered that active learning can improve the data efficiency of process supervision by 2.6 times. The authors have also released their comprehensive process supervision dataset, named PRM800K.

**Strengths:**

1.	This work offers an independent and thorough investigation into the superiority of outcome supervision versus process supervision methods, yielding new and valuable conclusions.

2.	The authors introduce an active learning method to enhance the data efficiency of process supervision, making it more practical for real-world applications.

3.	The authors have released their extensive human-labeled dataset.

**Weaknesses:**

The experiments were conducted solely on examination datasets. For more robust conclusions, it would be beneficial to conduct experiments on a broader range of significantly diverse datasets.

**Questions:**

Do the authors have more results on significantly different datasets?

---

> ### Author Response · Authors · 2023-11-14
>
> Thank you for the thoughtful comments. We agree that it is worth exploring this method in more general settings. We provide some evidence of generalization beyond math in Table 1 (to chemistry and physics in particular), and we hope that future work will continue to study this further.

---

### Official Review · Reviewer_4gd2 · 2023-10-31

**Soundness:** 3 good
**Presentation:** 3 good
**Contribution:** 2 fair
**Rating:** 3
**Confidence:** 3

**Summary:**

The paper proposes to provide language models with intermediate supervision on the step level instead of providing a single reward based on the whole generation. They collect step-level feedback with human data-labelers in the math domain and compare outcome-supervised reward models and process-supervised reward models on this dataset. They demonstrate that providing step-level feedback can achieve better performance and also show that active learning can be used to lower the data labeling cost.

**Strengths:**

1. The idea of providing fine-grained feedback for learning models is intuitive and technically sound.
2. They collect a dataset PRM800K that contains step-level labels across different solutions to math problems, which is helpful for future research in this direction.
3. The paper is well-written and easy to follow.

**Weaknesses:**

1. It is a well-known problem in the RL community that providing dense rewards can be better than providing sparse rewards, and there is a research direction on reward shaping that is very relevant to this idea. Therefore, the idea of providing intermediate rewards for training can lack novelty.
2. The paper mainly conducts experiments in a math dataset, where it is easy to provide step-by-step intermediate rewards. However, there are many other tasks such as essay writing and story generation where it can be hard to provide such feedback. Therefore, it is unclear if the method is generalizable.
3. It is hard to draw insights from their experiments. It would be better to see the advantages of using step-level rewards in more dimensions.

**Questions:**

1. Under the same budget, is it better to provide step-level rewards on a small set or outcome-level rewards on a large set?

---

> ### Author Response · Authors · 2023-11-14
>
> Thanks for your thoughtful feedback. We appreciate your points, especially that process-level reward for language models should be explored more fully across a variety of domains, but disagree on your overall assessment of our contribution.
>
> You are correct that dense rewards are well studied in RL. However, the majority of LLM RL literature is still focused on sparse rewards. That our methods strongly improve on current baselines shows that reward shaping in this domain is understudied. We think it is important for performance and alignment of LLMs to investigate intermediate rewards further (hence our contribution of a large open-source dataset of dense human feedback).
>
> We agree with your second point that these methods should be explored on more general distributions. However, the purpose of this paper is to present a clear, compelling argument that researchers should explore process-level feedback further. For this, it is better to start with domains that are easier to evaluate.
>
>
> To the question: We explicitly argue in section 2.6 that our process-level labels are equivalent in cost to outcome-level labels (so under the same budget the step-level rewards are better), and figure 4 shows about how big a set of outcome-level labels would be needed to recover the performance of our process-level model.

---

### Official Review · Reviewer_HKpB · 2023-11-01

**Soundness:** 3 good
**Presentation:** 3 good
**Contribution:** 3 good
**Rating:** 5
**Confidence:** 4

**Summary:**

The authors release a new dataset with process supervision, which provides human feedback for each intermediate reasoning step. The authors label the complex reasoning steps for solving MATH dataset. The authors find that process supervision significantly outperforms outcome supervision, whether only human feedback on the final answer, for training models. The authors also find that active learning significantly improves the efficacy of process supervision.

**Strengths:**

1, A new dataset with humans verifying each reasoning step.

2. The process supervision is important and the labeled dataset is useful for further research on reasoning with math.

3. The authors have also done experiments with active learning to improve the efficacy.

**Weaknesses:**

1. The work only explores the math problem. It would be better to explore different tasks.

2. The authors haven't applied it to RLHF. It is still not clear how process supervision and outcome supervision affect the generation model performance. Actually, if outcome supervision is large and diverse enough, the model trained with outcome supervision can also do process supervision by feeding the reason path step by step.

**Questions:**

Conducting RLHF would be more impressive.

---

> ### Author Response · Authors · 2023-11-14
>
> We appreciate the feedback. We address your concerns below.
>
> 1. We agree there is much more work to be done exploring process supervision in other domains, but we felt the math domain provided a clear and compelling testbed for our core insights. It is important for scientific arguments to be clear and compelling so that we can incrementally build strong foundations for future knowledge.
>
> 2. The primary purpose of this work is to study how to train reliable reward models, which is an important problem in its own right. While we agree that there are interesting connections to RLHF that are worth investigating further, we leave that to future work.
>
>
> We agree (and argue exactly in section 7.1) that a large enough outcome supervision dataset should recover the performance of our process reward model. I am not certain this is a weakness. We believe this is actually the core contribution of the paper: It is expensive to do RLHF because human feedback is expensive, so we show you can do RLHF more cheaply by collecting more dense labels from humans.

---

### Official Review · Reviewer_DxcT · 2023-11-05

**Soundness:** 3 good
**Presentation:** 3 good
**Contribution:** 3 good
**Rating:** 6
**Confidence:** 3

**Summary:**

This paper investigates the effectiveness of process supervision compared to outcome supervision for training reliable reward models in large language models, focusing on the challenging MATH dataset. The authors conduct a detailed comparison of these two types of supervision for training reward models, using a more capable base model and significantly more human feedback.

The main contributions of the paper are
  * demonstrating that process supervision can train more reliable reward models than outcome supervision, with their model solving 78.2% of problems from a representative subset of the MATH test set
  * showing that active learning can lead to a 2.6× improvement in the data efficiency of process supervision
  * releasing their full process supervision dataset, PRM800K, to promote related research.

**Strengths:**

1. Assigning rewards for intermediate steps is a novel and intuitive idea. Compared to assigning rewards for the outcome alone, judging intermediate steps better leverages problem structures.
2. The empirical results are very strong.
3. The released dataset can support the research community to further explore this direction.

**Weaknesses:**

1. The reproducibility for this work is concerning. It is hard to understand under what conditions one can successfully train an effective process reward model.  The authors did not have sufficient details for both models and data.  In the paper, the authors stated, "The small-scale base models are similar in design to GPT-4, but they were pretrained with roughly 200 times less compute." However, the paper neither reveals the size of the small-scale base models nor the large-scale base models.  It is unclear whether the data gets the PRM to work or the scale of the model that gets it to work. I understand that work done at industry has its confidentiality rules to follow, but the authors can maybe conduct experiments on open models on their pretraining and fine-tuning datasets? Due to the reproducibility issue, I think this work would be a better fit for a blog post than a main conference paper.
2. It is unclear whether PRM would work for other types of reasoning problems.

**Questions:**

1. Who are the human data-labelers?  Are they MTURK crowdsource workers, math teachers, or? What is their proficiency level?
2. "MathMix consists of a smaller set of 1.5B tokens containing individual math problems and their solutions" ---> where are these math problems from?  textbooks or scraping the web?

---

> ### Author Response · Authors · 2023-11-14
>
> Thank you for the careful review of our paper. We think this review accurately identifies the contributions and limitations of our work.
>
> We should absolutely continue exploring the generalization of our methods into other domains, but believe it is reasonable to begin with math. We hope our generalization results in Table 1 (into chemistry and physics in particular) give some initial evidence that this method generalizes.
>
> We also appreciate the concerns around reproducibility. While we believe our small-scale results should be wholly reproducible on open-source models, it would have been more responsible of us to test that directly. The large-scale results are likely not replicable on any currently-available open-source models. This is an unfortunate limitation of the current state of the technology and a valid criticism. We still believe it’s important that we report results at the frontiers of scale. We think this is a valid contribution to the literature, since our results suggest that process supervision enables more cost-efficient reward model training at all model sizes.

---

### Meta-Review · Area_Chair_9Dqd · 2023-12-13

**Metareview:**

**Paper Summary:**

This paper studies the effectiveness of process supervision compared to outcome supervision by providing step-level human feedback to train models. It also shows that active learning can lead to an improvement in the data efficiency of process supervision. The release of the dataset PRM800K is an additional contribution, which might benefit further research in this area.

**Strengths:**

1. Intuitive Approach: Assigning rewards for intermediate steps, as opposed to solely the outcome, is an intuitive way of leveraging problem structures for training models (DxcT, 4gd2, HKpB).
2. Strong Empirical Results (DxcT).
3. Dataset Release: The release of PRM800K offers a valuable resource for the community to explore similar research directions (DxcT, HKpB, 4gd2, 5YZq).

**Weaknesses:**

1. Reproducibility Concerns: There are significant concerns regarding the reproducibility of the work, due to insufficient details about the models and data used in the experiments (DxcT).
2. Limited Scope of Experiments: The work focuses on the math problem domain, raising questions about the generalizability of the findings to other types of reasoning tasks (DxcT, HKpB, 4gd2, 5YZq).

**Decision:**

Despite the noted weaknesses, particularly concerning reproducibility and the scope of experimentation, the paper stands out for its novel approach to training, strong empirical results, and dataset contribution. Therefore, I recommend the acceptance of this work.

**Justification For Why Not Higher Score:**

Several reviewers have raised concerns about reproducibility, and the applicability of this approach to other problems.

**Justification For Why Not Lower Score:**

The approach of process supervision is intuitive and the results are strong.

---

### Decision · Program_Chairs · 2024-01-16

Accept (poster)